# A Novel Multiprotein Bridging Factor 1-Like Protein Induces Cyst Wall Protein Gene Expression and Cyst Differentiation in *Giardia lamblia*

**DOI:** 10.3390/ijms22031370

**Published:** 2021-01-29

**Authors:** Shao-Wei Huang, Zi-Qi Lin, Szu-Yu Tung, Li-Hsin Su, Chun-Che Ho, Gilbert Aaron Lee, Chin-Hung Sun

**Affiliations:** 1Department of Tropical Medicine and Parasitology, College of Medicine, National Taiwan University, Taipei 100, Taiwan; q623280@hotmail.com (S.-W.H.); 100312015@gms.tcu.edu.tw (Z.-Q.L.); sayeodong@gmail.com (S.-Y.T.); hsin93@yahoo.com.tw (L.-H.S.); f123ens@yahoo.com.tw (C.-C.H.); 2Department of Medical Research, Taipei Medical University Hospital, Taipei 110, Taiwan; gilbertlee@mail.knu.edu.tw

**Keywords:** cyst, MBF1, transcription, *Giardia*, differentiation, DNA binding protein, parasite, transcription regulation

## Abstract

The capacity to synthesize a protective cyst wall is critical for infectivity of *Giardia lamblia*. It is of interest to know the mechanism of coordinated synthesis of three cyst wall proteins (CWPs) during encystation, a differentiation process. Multiprotein bridging factor 1 (MBF1) gene family is a group of transcription coactivators that bridge various transcription factors. They are involved in cell growth and differentiation in yeast and animals, or in stress response in fungi and plants. We asked whether *Giardia* has MBF1-like genes and whether their products influence gene expression. BLAST searches of the *Giardia* genome database identified one gene encoding a putative MBF1 protein with a helix-turn-helix domain. We found that it can specifically bind to the AT-rich initiator promoters of the encystation-induced cwp1-3 and myb2 genes. MBF1 localized to cell nuclei and cytoplasm with higher expression during encystation. In addition, overexpression of MBF1 induced cwp1-3 and myb2 gene expression and cyst generation. Mutation of the helixes in the helix-turn-helix domain reduced cwp1-3 and myb2 gene expression and cyst generation. Chromatin immunoprecipitation assays confirmed the binding of MBF1 to the promoters with its binding sites in vivo. We also found that MBF1 can interact with E2F1, Pax2, WRKY, and Myb2 transcription factors that coordinately up-regulate the cwp genes during encystation. Using a CRISPR/Cas9 system for targeted disruption of mbf1 gene, we found a downregulation of cwp1-3 and myb2 genes and decrease of cyst generation. Our results suggest that MBF1 is functionally conserved and positively regulates *Giardia* cyst differentiation.

## 1. Introduction

*Giardia lamblia* parasitizes the human small intestine to cause diarrheal disease worldwide [1,2]. Poor hygiene and water resource contamination can result in the transmission of giardiasis [1,3,4]. Chronic diarrhea due to giardiasis may lead to malnutrition and growth failure in children [5,6,7]. Recently, *Giardia* infection has been recognized as a cause of higher risk of irritable bowel syndrome with persisting abdominal symptoms [8,9].

As a single-cell protozoan, *Giardia* is a valuable model for understanding the evolution of cell differentiation [1]. The presence of fewer basic components for biological pathway suggest that *Giardia* exhibits unusual mechanisms as compared with other eukaryotes [10]. Like many other protozoa that persist in a dormant state, *Giardia* differentiates from a pathogenic trophozoite into a resistant walled cyst, which is a tactic for resistance of hypotonic lysis in fresh water and survival in gastric acid during transmission [1,3]. Proteins and polysaccharides are the major components of the cyst wall, whose encystation-specific synthesis is key to encystation [11,12,13]. During encystation, three cyst wall proteins (CWPs) are newly synthesized to form cyst wall [14,15,16]. It is of interest to identify factors involved in up-regulation of cwp genes. Several transcription factors are induced in cyst differentiation, including Myb2 (Myb1-like protein in the *Giardia* genome database), GARP1, ARID1, WRKY, E2F1, Pax1, and Pax2 and participate in this process [17,18,19,20,21,22,23,24].

Bacteriophage Cro and CI (λ) repressors contains typical helix-turn-helix (HTH) domains of about 50–60 residues [25,26]. The basic core of HTH domain contains three α-helices (H1, H2 and H3) separated by two short turns [25,26]. H2 and H3 are critical for DNA interaction [25,26]. H3 is the recognition helix that is needed for DNA binding and recognizes specific sequence along the DNA major groove [25,26]. H2 can help nonspecifically stabilize the complex [25,26]. H1 has a slight role in DNA binding. The Cro/CI-type HTH domain are present in many transcription regulators of prokaryotes and eukaryotes [25,26], including multiprotein bridging factor 1 (MBF1). MBF1 proteins contain a highly conserved HTH domain with four α-helixes [27,28]. MBF1 may have DNA binding activity because HTH domain is a typical DNA binding domain [27,28]. MBF1 has not been found in bacteria [27]. The archaea MBF1 proteins are quite different from the eukaryotic ones as they have an extra Zn ribbon motif [27].

MBF1 was named because it functioned as a transcriptional coactivator by bridging regulatory transcription factors and general transcription factors in higher eukaryotes [29]. MBF1 proteins have been found in yeast and human and play important roles in regulating cell proliferation and differentiation [29,30,31,32]. MBF1 is a co-activator that bridges GCN4, a transcriptional activator of the bZIP family that regulates amino acid biosynthesis in yeast [32]. MBF1 is also named as endothelial differentiation-related factor 1 (EDF1) as it plays a role in differentiation of epithelial cells in human [33]. MBF1 interacts with a sequence specific DNA binding factor of steroid receptor family, fushi tarazu factor 1 (FTZ-F1), to induce transcription of fushi tarazu gene which encodes a product working in *Drosophila* early embryo segmentation [34]. MBF1 interacts with a general transcription factor, TATA-binding protein, and a gene-specific transcription factor, FTZ-F1, for FTZ gene transactivation by stabilizing the protein-DNA interactions in development of silkworm *Bombyx mori* [35]. In addition, MBF1 gene was up-regulated by a virus that causes cell death of lobsters, suggesting that MBF1 is stress-related [36,37]. MBF1 functions as a positive modulator of AP-1 for its DNA binding by preventing an oxidative modification of AP-1 during *Drosophila* oxidative stress [38]. MBF1 may be involved in stress tolerance and in ethylene signal transduction pathway for seed germination in *Arabidopsis* [39]. MBF1 also plays a role in stress response in fungi [40]. In another aspect, MBF1 proteins play roles in translation process in archaea and yeast [41,42].

Little is known about MBF1 in protozoan parasites. To date, only *Cryptosporidium* MBF1 has been characterized [43]. It is a transcription co-activator working with TATA-binding protein [43]. Because of the critical roles of MBF1 proteins in cell proliferation and cell differentiation of many eukaryotes, we asked whether *Giardia* has MBF1 family proteins and whether they function in inducing *Giardia* differentiation into dormant cysts. We searched the *Giardia* genome database and identified one MBF1 homolog (MBF1). We found that MBF1 was up-regulated during encystation and it possessed DNA-binding activity to AT-rich initiator sequences. We also found that MBF1 can interact with important encystation-induced E2F1, Pax2, WRKY, and Myb2 transcription factors. Using overexpression and mutation analysis, and a CRISPR/Cas9 system [44] for targeted disruption of the mbf1 gene, we found that MBF1 can up-regulate cwp1-3 and myb2 genes and induce cyst generation. Our results suggest that MBF1 may be an important transcription factor promoting *Giardia* differentiation.

## 2. Results

### 2.1. Analysis of the mbf1 Gene

An MBF1 homolog (GenBank accession number XP_001704109.1, open reading frame 2732) was identified using a BLAST search of the *G. lamblia* genome database using the sequence of human MBF1 (GenBank accession number NP_003783.1) as a query [10,45]. The deduced *Giardia* MBF1 protein is composed of 107 amino acids with a pI of 10.22 and a molecular weight of 11.97 kDa. Like human MBF1, the *Giardia* MBF1 has a putative 4-helix HTH DNA-binding domain near C-terminus as predicted by Pfam (Appendix A and Figure 1A) [29,46].

The sequence of the HTH domain of the *Giardia* MBF1 has some similarity to those of the MBF1 family from other eukaryotes (Figure 1B). The λ CI repressor was used as a model for alignment of the HTH domain [47]. The predicted secondary structure of the HTH domain of *Giardia* MBF1 suggests similar 4-helix HTH structure (Psipred prediction) as compared to the HTH domains of human and other eukaryotic MBF1 proteins (Appendix A and Figure 1B) [48]. The *Giardia* MBF1 has 16.11% sequence identity and 29.53% sequence similarity to the human MBF1. Few of the key contact residues identified by structural studies of λ CI repressor are conserved in *Giardia* MBF1 (Figure 1B). The HTH domain of λ CI repressor is composed of four helixes, and helix 3 is the recognition helix (Figure 1B) [49]. The MBF1 protein from human, *Drosophila*, *Arabidopsis*, or *Dictyostelium* contained a MBF domain near the N terminus as predicted by Pfam (Appendix A). These MBF1 domains contained many basic amino acids [27]. The N-terminal region of *Giardia* MBF1 contained several basic amino acids (Appendix A). *Giardia* MBF1 does not have the MBF1 domain as predicted by pfam (Appendix A) [46].

### 2.2. Encystation-Induced Expression and Localization of the MBF1 Protein

We found that the *mbf1* mRNA increased by ~1.55-fold in 24 h encysting cells using RT-PCR and quantitative real-time PCR analysis (Figure 2A). To investigate the expression of MBF1 protein, we generated an antibody specific for the full-length MBF1. Western blot analysis confirmed that this antibody recognized MBF1 at a size of ~12 kDa (Figure 2B). Interestingly, MBF1 level significantly increased during encystation (Figure 2B).

To overexpress the MBF1 protein, we prepared construct pPMBF1, in which the mbf1 gene is directed by its own promoter (Figure 2C). The HA tag (~1 kDa) was also predicted to fuse to the C terminus of the protein (Figure 2C). After stable transfection with the plasmid, we found the level of the MBF1-HA protein significantly increased during encystation (Figure 2D), which was similar to the expression pattern of the endogenous MBF1 protein (Figure 2B). We also found that the epitope-tagged MBF1-HA protein was localized to both the nuclei and cytoplasm during both vegetative growth and encystation stages (Figure 2E), indicating that MBF1 could go into the nuclei in *Giardia*.

### 2.3. MBF1 Can Induce the Expression of the cwp1-3, and myb2 Genes

We further determined the effect of the *Giardia* MBF1 on encystation-induced genes. We confirmed the overexpression of the MBF1 protein and found a significant increase in the cyst wall protein 1 (CWP1) protein level in the MBF1-overexpressing cell line relative to the vector control cell line (Figure 2C and Figure 3A) [50]. We found a significant increase in the mRNA expression of the endogenous mbf1 plus vector-expressed mbf1 in the MBF1-overexpressing cell line in comparison with the control cell line using RT-PCR and quantitative real-time PCR analysis (Figure 3B,C). Interestingly, the mRNA levels of the cwp1-3 and myb2 genes were also significantly increased in the MBF1-overexpressing cell line (Figure 3B,C). In addition, we also found a significant increase in the cyst number in the MBF1-overexpressing cell line (Figure 3D). Similar results were observed in encystation samples (Appendix A). The results suggest that overexpression of MBF1 could induce the cwp1-3 and myb2 gene expression and cyst generation.

### 2.4. Effect of the MBF1 Mutants

We also performed mutation analysis to understand the role of MBF1. We found that mutation of the H3 (MBF1m1, residues 70–79), H2 (MBF1m2, residues 60–69), and H4 (MBF1m3, residues 87–96), did not change the localization compared to the wild-type MBF1 (Figure 1B and Figure 2E, and Appendix A). We also found that mutation of H4 and the turn between H3 and H4 (MBF1m4, residues 82–91) did not change the localization (Figure 1B and Appendix A). Mutation of these residues disrupted the formation of α-helixes as predicted by secondary structure prediction analysis (Appendix A) (http://www.bioinf.manchester.ac.uk/dbbrowser/bioactivity/NPS2.html).

We further analyzed the effect of mutation of MBF1. We found that the protein level of MBF1m2 was similar to that of the wild-type MBF1, but the levels of MBF1m1, MBF1m3, and MBF1m4 significantly decreased (Figure 3A and Appendix A). We also found that the CWP1 level significantly decreased in the MBF1m1-, MBF1m2-, MBFm3-, and MBF1m4- expressing cell lines relative to the wild-type MBF1-expressing cell line (Figure 3A). We further measured ed the mRNA levels in different transfectants. As shown by RT-PCR and quantitative real-time PCR analysis, the mRNA levels of mbf1m1-HA, mbf1m3-HA, and mbf1m4-HA significantly decreased compared with that of wild type mbf1-HA during vegetative growth, but the mRNA level of mbf1m2-HA was similar to that of the wild-type mbf1-HA (Figure 3B,C, and Appendix A). We did not detect any mbf1-HA transcripts in the control cell line (Figure 3B). Interestingly, we found a significant reduction in the mRNA levels of cwp1-3 and myb2 in the MBF1m1-, MBF1m2-, MBF1m3-, and MBF1m4-expressing cell lines in comparison with the wild-type MBF1-expressing cell line (Figure 3B,C and Appendix A). In addition, we also found a significant reduction in cyst number in the MBF1m1-, MBF1m2-, MBF1m3-, and MBF1m4- expressing cell lines in comparison with the wild type MBF1-expressing cell line (Figure 3D and Appendix A). We also obtained similar results in encystation samples (Appendix A). The results suggest a decrease of transactivation and encystation-inducing activity of MBF1m1, MBF1m2, MBF1m3, and MBF1m4.

Oligonucleotide microarray assays suggest that 30 and 23 genes were significantly up-regulated (≥2-fold) and down-regulated (≤1/2) (*p* < 0.05) in the MBF1-overexpressing cell line relative to the vector control cell line, respectively (Appendix A). Interestingly, *e2f1* gene encoding a transcription factor involved in inducing cwp gene expression was found to be up-regulated by MBF1 overexpression (Appendix A) [23].

### 2.5. MBF1 Has DNA Binding Activity

The Cro/CI type HTH proteins have the ability to bind DNA [25,26]. *Arabidopsis* MBF1 can bind to the specific CTAGA element [51]. To test DNA binding activity of MBF1, we expressed MBF1 in *E. coli* and purified it to >95% homogeneity. We performed electrophoretic mobility shift assays with purified recombinant MBF1 and double-stranded DNA sequences from the 5′-flanking region of an encystation-induced gene, cwp1 [14]. Using a labeled double-stranded DNA probe, cwp1-45/-1, and MBF1, we showed the formation of shifted bands (Figure 4A). The cwp1-45/-1 probe represents the -45/-1 region of the cwp1 promoter which contains an AT-rich initiator (Inr) (Figure 4B). We further confirmed the binding specificity using supershift and competition assays (Figure 4A). The retarded cwp1-45/-1 bands were supershifted by an anti-V5-horseradish peroxidase (HRP) antibody (Figure 4A). The retarded cwp1-45/-1 bands can be competed out by a 200-fold molar excess of unlabeled cwp1-45/-1 probe (Figure 4A). As non-specific competition, the same excess of the cold oligonucleotide bearing the GC-rich 18S-30/-1 did not reduce the intensity of the retarded bands (Figure 4A). The results suggest that MBF1 can bind to the cwp1 promoter containing AT-rich Inr, but not to GC-rich sequence. We further tested whether MBF1 binds to the 5′ region of other encystation-induced genes, cwp2, cwp3, and myb2 [15,16,17], and found that MBF1 also bound to the cwp2-60/-31, cwp2-30/+8, cwp3-60/-31, cwp3-30/+10, and myb2-30/-1 probes (Figure 4B). In addition, MBF1 bound to a well characterized ran core AT-rich promoter, ran-51/-20, and weakly to ran-30/-1, but not to the 18S-30/-1, probe (Figure 4B) [52]. MBF1 also bound to the cwp1-90/-46 probe (Figure 5A). Interestingly, we further found that MBF1 bound to specific AT-rich sequences, including a poly(A) sequence and a poly(A) sequence with a T, or TT insertion (Figure 5A). Our findings suggest that MBF1 can bind to the AT-rich promoter regions of the cwp1, cwp2, cwp3, myb2, and ran genes.

It has been shown that the Cro/CI repressor can bind to both DNA major and minor grooves [53,54]. To further understand how MBF1 binds DNA, we used distamycin A, which binds to the minor groove of AT-rich DNA sequences, as a competitive inhibitor of MBF1 binding [55]. As shown in Figure 5B, the binding of MBF1 to DNA decreased with increasing concentrations of distamycin A. We also used methyl green, a DNA major-groove binding drug, as a competitive inhibitor of MBF1 binding, and found that the binding of MBF1 to DNA decreased with increasing concentrations of methyl green (Figure 5C) [56].

### 2.6. Analysis of DNA Binding Activity of MBF1 Mutants

To understand which regions are important for MBF1 binding to DNA, we performed DNA binding assays with purified recombinant MBF1 mutants (Figure 6A). As shown in Figure 6B and Appendix A, there was no change of the DNA binding activity of MBF1m1, MBF1m3, and MBF1m4. We found an obvious decrease of DNA binding activity of MBF1m2 (Figure 6B), suggesting that helix 2 is important for DNA binding (Figure 1B).

### 2.7. In Vivo Association of MBF1 with the cwp1-3, myb2, and mbf1 Its Own Promoters

We further used chromatin immunoprecipitation (ChIP) assays to investigate the interaction of MBF1 with specific promoters in the MBF1-overexpressing cell line in vivo. We found that MBF1 associated with multiple regions, including the cwp1-3 and myb2 gene promoters and its own promoter, during encystation (Figure 6C,D). However, MBF1 did not interact with the promoters without the MBF1 binding sites, including the promoter of the U6 snRNA gene which is transcribed by RNA polymerase III, or the promoter of the 18S ribosomal RNA gene which is transcribed by RNA polymerase I (Figure 6C,E).

### 2.8. Interaction of MBF1 and E2F1, Pax2, or WRKY in a Complex

MBF1 promotes cell proliferation and differentiation by interacting with other transcription activators in yeast, human, or fly [29,30,31,32,34]. Therefore, *Giardia* MBF1 may regulate genes by interacting with other classes of transcription factors. We further tried to understand whether MBF1 can interact with E2F1, the encystation-induced transcription factor involved in up-regulation of the cwp genes during encystation [23]. We used the E2F1-overexpressing cell line to analyze the interaction between these two transcription factors (Figure 7A,B) [23]. The pPE2F1 stable cell line expressed the E2F1-HA protein, in but the vector control cell line did not (Figure 7A,B). We found that E2F1-overexpression resulted in an increase in the MBF1 protein level (Figure 7B), but led to a decrease in the ISCS level (Figure 7B). The HA-tagged E2F1 protein was immunoprecipitated from cell lysate with anti-HA antibody and Western blots were probed with anti-HA and anti-MBF1 antibodies. The results indicated that MBF1 co-precipitated with E2F1-HA (Figure 7C). As a negative control, no anti-HA or anti-MBF1 antibody reactivity was obtained with immunoblots of anti-HA immunoprecipitates of the control cell line not expressing E2F1-HA (Figure 7C). No anti-ISCS antibody reactivity was obtained with immunoblots of anti-HA immunoprecipitates of the pPE2F1 cell line (Figure 7C). The reciprocal immunoprecipitation also confirmed the interaction of E2F1 and MBF1 (Figure 7D), suggesting that E2F1 and MBF1 were present in a complex. Similarly, we also found an interaction between Pax2 (Figure 7E–G), WRKY (Figure 7H–J), or Myb2 (Figure 7K–M) and MBF1 in a complex.

### 2.9. Reduced Expression of cwp1-3 and myb2 Genes by Targeted Disruption of the mbf1 Gene

We have developed a CRISPR/Cas9 system to successfully disrupt the mlf and top3β genes [44,57]. We further used this system for the targeted disruption of the mbf1 gene. After transfection of the CRISPR/Cas9 constructs into *Giardia*, the MBF1td stable transfectants were established under puromycin selection (Figure 8A). We confirmed the replacement of the mbf1 gene with the puromycin acetyltransferase (pac) gene by PCR and sequencing analysis of genomic DNA (Figure 8B,C and Appendix A), and found a successful disruption of the mbf1 gene by about 29% and a partial replacement of the mbf1 gene with the pac gene (Figure 8B,C). G418, an inhibitor of polypeptide synthesis, had cytotoxicity and was used to test the drug sensitivity of the top3β targeted disruption cell line [57]. We also used G418 to test the drug sensitivity of the MBF1td cell line and found that the viability of the MBF1td cell line decreased compared to the control cell line after G418 treatment, suggesting that MBF1td cell line exhibited increased sensitivity to G418 (Figure 8D). In addition, we found a significant decrease in cyst number in the MBF1td cell line in comparison with the control cell line during vegetative growth (Figure 8E).

We further found a significant decrease in MBF1 and CWP1 protein levels in the MBF1td cell line in comparison with the control cell line (Figure 8F). In addition, we also found a significant reduction in the mRNA levels of mbf1, cwp1-3, or myb2 in the MBF1td cell line in comparison with the control cell line (Figure 8G). We also found that the cyst wall is thinner in the MBF1td cell line relative to the control cell line (Figure 8H), correlating with the relative expression levels of cwp1-3 genes (Figure 8F,G). Similar results were obtained from the MBF1td cell line during encystation (Appendix A). We further performed analysis after removal of puromycin and found that similar results were obtained from the MBF1td –pu cell line during both vegetataive growth and encystation (Appendix A). Our results indicate that targeted disruption of the mbf1 gene resulted a decrease in expression of cwp1-3 and myb2, drug sensitivity, and cyst generation, suggesting a positive role of MBF1 in encystation.

### 2.10. G418 and Curcumin Increased the Levels of MBF1 and CWP1 Proteins

MBF1 family proteins are involved in stress response in many organisms, including plants and fungi [36,37,38,39,40]. We tried to understand whether MBF1 was stress-induced using used two stress-related drugs, G418 and curcumin, which are well-known inducers of reactive oxygen species generation and contributes to the oxidative stress [58,59,60]. It has been shown that both drugs have cytotoxic effect on *Giardia* [61,62]. We found that G418 and curcumin treatment significantly increased the levels of MBF1 and CWP1 proteins (Figure 9A,B). The results suggested that *Giardia* MBF1 was stress-related, which is similar to the up-regulation of MBF1 by stress response in other organisms [36,37,38,39,40].

## 3. Discussion

MBF1 homologues are transcription factors important in a variety of key cellular functions including regulation of transcription, stress response, and cell proliferation and differentiation [29,30,31,32,39]. In this study, we identified and characterized a putative MBF1 transcription factor from *Giardia*. We found that *Giardia* MBF1 had a similar function in inducing transcription for cyst differentiation and it was stress-related, suggesting that the eukaryotic MBF1 family may have evolved before divergence of *Giardia* from the major eukaryotic line of descent.

MBF1 proteins in higher eukaryotes are involved in differentiation and function as transcriptional activators [29,30,31]. They regulate specific target genes by interacting with other classes of DNA-binding proteins [34]. For example, *Drosophila* MBF1 interacts with nuclear receptor fushi tarazu factor 1 as a cofactor for inducing embryo development [34]. We found that *Giardia* MBF1 protein was expressed at a higher level during stress response and encystation (Figure 2B,D and Figure 9A,B). MBF1 has DNA binding activity for the core AT-rich Inr promoters of the cwp1-3 and myb2 genes (Figure 4). We hypothesize that MBF1 and other transcription factors that bind to the AT-rich elements or the proximal upstream regions can form complexes (Figure 10). RNA polymerase II was then recruited by the complex to activate cwp transcription (Figure 10). It is possible that interaction of MBF1 with some of these transcription factors can stabilize their binding to DNA. A similar phenomenon was reported for the MBF1–FTZ-F1 interaction in silkworm [35].

Our results showed that overexpressed MBF1 increased the expression of cwp1-3 genes by ~2.22–3. 91-fold in vegetative trophozoites (Figure 3C). However, the cwp1 promoter could be increased by ~47-fold during encystation [18], suggesting that *Giardia* MBF1 may need to cooperate with some encystation-induced cofactors on the promoter context of encystation-induced genes. *Giardia* MBF1 may directly bind and regulate gene promoter, but it may also interact with other transcription factors, such as E2F1, Pax2, WRKY, or Myb2, to regulate the cwp gene expression (Figure 7). MBF1 can bind to AT-rich elements of the constitutive ran gene (Figure 5B). However, the overexpressed MBF1 did not increase the ran gene expression (Figure 3A–C), possibly due to the absence of cooperative transactivation of the constitutive ran gene by the encystation-induced transcription factors.

Many *Giardia* gene promoters, including the encystation-induced cwp promoters, are short and contain AT-rich Inr elements responsible for promoter activity and transcription start site selection [14,15,16,52,63,64]. To date, we have identified seven families of transcription factors in *Giardia* that are up-regulated during encystation and involved in the transactivation of the cwp genes encystation, including Myb, GARP, ARID, WRKY, Pax, E2F, and MBF1 [17,18,19,20,21,22,23,24]. It is interesting that *Giardia* possesses the transcription factors identified in both plants and animals (Myb, ARID, E2F, and MBF1) or the transcription factors identified only in plants (GARP and WRKY) and animals (Pax). Among them, ARID1, Pax, and MBF1 can bind AT-rich Inr elements [19,22,24]. Mutation of the AT-rich Inr element in the cwp1 promoter might disrupt the binding of these proteins to the Inr, leading to a downstream shift in transcription start site selection [19]. The proximal upstream regions with Myb2, GARP, WRKY or E2F1 binding sites are just next to the AT-rich Inr elements [17,18,19,20,21,23], making it easy for regulatory and general transcription factors to interact with each other. In this study, we also found that MBF1 can also bind to the AT-rich promoters of the cwp genes in vitro (Figure 4). ChIP assays also confirmed the association of MBF1 with its own promoter and the cwp1-3 and myb2 promoters in vivo (Figure 6). In addition, MBF1 can be co-immunoprecipitated with E2F1, Pax2, WRKY, or Myb2 (Figure 7). The interaction between MBF1 and these transcription factors may be required for promoter activity and accurate transcription start site selection.

The HTH domain of *Giardia* MBF1 has few of the conserved key contact residues, and it has a predicted helix-turn-helix structure similar to that of other MBF1 family members (Figure 1B and Appendix A). *Arabidopsis* MBF1 may bind to the specific CTAGA element to activate expression of 36 genes during heat stress [51]. KdpE, a winged helix-turn-helix transcription factor, binds AT-rich sequence and positively regulates operon to express the potassium ion channel in *Escherichia coli* [65]. The basic residues R193 and R200 in the recognition helix are important for DNA binding of KdpE [65]. Interestingly, we also found that mutation of the helix 2 of *Giardia* MBF1, including two basic residues, R60 and K61, resulted a decrease of DNA binding activity (Figure 1B and Figure 6B). In addition, *Giardia* MBF1 can bind to the AT-rich promoter regions of cwp1-3 and myb2 genes (Figure 4). Further studies also indicate that MBF1 can bind to poly(A) sequence with a T or TT insertion, but not to the GC-rich sequence (Figure 4A and Figure 5A), suggesting that *Giardia* MBF1 may recognize variable AT-rich Inr sequences in different gene promoters. The increase of chemosensitivity to G418 in the MBF1td cell line (Figure 8D) suggests that MBF1 may affect many genes important for cell growth and for survival in antibiotic stress.

The helix 3 of CI repressor is the recognition helix, which corresponds to the helix 3 of MBF1 (Figure 1B). However, mutation of the helix 3, helix 4, or the turn between the helix 3 and helix 4 of *Giardia* MBF1, did not affect DNA binding, although with disruption of α-helix structure (MBF1m1, m3, or m4) (Figure 6B; Appendix A). The presence of another α-helix near *Giardia* MBF1 N terminus (N-terminal helix) suggests that the helix 2 could play a role of helix 3 as the recognition helix (Appendix A). Interestingly, mutation of the helix 2 of MBF1 decreased DNA binding (Figure 6B).

Our results suggest that the HTH domain is not only essential for DNA binding, but also important for in vivo function. We found that mutation of the helix 3 (MBF1m1) or the helix 4 (MBF1m3 or MBF1m4) did not affect DNA binding but resulted in an obvious decrease of the levels of CWP1 protein, cyst generation, cwp1-3 and myb2 mRNA (Figure 3A–D, Figure 6B and Appendix A). This indicates that these specific regions of the HTH domain may be positive regulatory regions for activation of transcription. It has been shown that residues 24–69 of MBF2, which span some part of the HTH domain, is important for interaction with TFIIA in *Drosophila* [34]. On the other hand, mutation of the helix 2 (MBF1m2) also resulted in a significant decrease of the levels of CWP1 protein, cyst generation, and cwp1-3 and myb2 mRNA (Figure 3A–D), and the effect is similar to the MBF1m1 and MBF1m3. Because MBF1m2 was expressed at higher levels than MBF1m1 and MBF1m3, its low inducing activity for the cwp genes may be due to its inability to bind DNA (Figure 3A and Figure 6B). Our results suggest that the HTH domain may be important for DNA binding and activation of transcription.

*Giardia* MBF1 has no typical nuclear localization signal as predicted with the PSORT software [66]. It is possible that passive diffusion allows transport of proteins of <20 kDa between nucleus and cytoplasm [67]. The presence of *Giardia* MBF1 in both nuclei and cytoplasm could be due to its small size (11.97 kDa) (Figure 2E). Mutation of either helix 2, 3, or 4 (MBF1m1, MBF1m2, MBF1m3, or MBF1m4) did not affect the localization of MBF1 to the nuclei and cytoplasm (Appendix A). MBF1 did not change its localization during encystation (Figure 2E). Similarly, transcriptional induction function of MBF1 is not controlled by nuclear translocation in *Arabidopsis* [37].

Our study provides evidence for the involvement of MBF1 in DNA binding and induction of the cwp1-3 and myb2 gene expression and cyst generation in the protozoan *G. lamblia*, suggesting that MBF1 may be functionally conserved and play a role in gene transcription and cell differentiation. Our studies provide new insights into the evolution of eukaryotic DNA binding domain from primitive to more complex eukaryotic cells and into the distinct function of MBF1 in differentiation of *Giardia* trophozoites into cysts.

## 4. Materials and Methods

### 4.1. G. Lamblia Culture

Trophozoites of *G. lamblia* WB, clone C6 (see ATCC 50803) (obtained from ATCC), were cultured in modified TYI-S33 medium [68]. Encystation was performed as previously described [16]. In experiments exposing *Giardia* vegetative trophozoites to G418 or curcumin, MBF1td and control cell line were cultured in growth medium at a beginning density of 1 × 10^6^ cells/mL with 217 μM or 518 μM G418 or 200 µM curcumin.

### 4.2. Cyst Count

Cyst count was performed on the stationary phase cultures (~2 × 10^6^ cells/mL) during vegetative growth as previously described [61]. Cyst count was also performed on 24 h encysting cultures. Total cysts including both type I and II cysts [69] were counted in a hemacytometer chamber.

### 4.3. Isolation and Analysis of the mbf1 Gene

Synthetic oligonucleotides used are shown in Appendix A. The *G. lamblia* genome database [10] was searched was searched with the amino acid sequence of the human MBF1 (GenBank accession number NP_003783.1) using the BLAST program [45]. This search detected one putative homologue for MBF1 (GenBank accession number XP_001704109.1, open reading frame 2732 in the *G. lamblia* genome database). The MBF1 coding region with 300 bp of 5′- flanking region was cloned and sequenced. We also performed RT-PCR with mbf1-specific primers using total RNA from *G. lamblia* to isolate its cDNA as previously described [22]. The cDNA was used as a template in subsequent PCR with primers mbf1F and mbf1R. Genomic and RT-PCR products were cloned into pGEM-T easy vector (Promega) and sequenced (Applied Biosystems, ABI, Foster City, CA, USA) and the results indicated no introns in the mbf1 gene.

### 4.4. Genomic DNA Extraction, PCR and Quantitative Real-Time PCR Analysis

Synthetic oligonucleotides used are shown in Appendix A. Genomic DNA was isolated from trophozoites using standard procedures as previously described [44,70]. PCR analysis of mbf1 (XP_001704109.1, orf 2732), cwp1 (U09330, orf 5638), cwp2 (U28965, orf 5435), and ran (U02589, orf 15869) genes was performed using primers mbf1F (PCR1F) and mbf1R (PCR1R), PCR2F and PCR2R, cwp1F and cwp1R, cwp2F and cwp2R, ranF and ranR, respectively. Quantitative real-time PCR was conducted as previously described [44]. Specific primers were designed for detection of the mbf1, cwp1, cwp2, and ran genes: mbf1realF and mbf1realR; cwp1realF and cwp1realR; cwp2realF and cwp2realR; ranrealF and ranrealR.

### 4.5. RNA Extraction, RT-PCR and Quantitative Real-Time PCR Analysis

Synthetic oligonucleotides used are shown in Appendix A. Total RNA was extracted and RT-PCR was performed as previously described [44]. The cDNA was used as a template in subsequent PCR. Semi-quantitative RT-PCR analysis of mbf1 (XP_001704109.1, orf 2732), mbf1-ha, cwp1 (U09330, orf 5638), cwp2 (U28965, orf 5435), cwp3 (AY061927, orf 2421), myb2 (AY082882, orf 8722), ran (U02589, orf 15869), and 18S ribosomal RNA (M54878, orf r0019) gene expression was performed using primers mbf1F and mbf1R, mbf1F and HAR, cwp1F and cwp1R, cwp2F and cwp2R, cwp3F and cwp3R, myb2F and myb2R, ranF and ranR, 18SrealF and 18SrealR, respectively. Quantitative real-time PCR was performed as previously described with specific primers for detection of the mbf1, mbf1-ha, cwp1, cwp2, cwp3, myb2, ran, and 18S ribosomal RNA genes: mbf1realF and mbf1realR; mbf1F and HAR; cwp1realF and cwp1realR; cwp2realF and cwp2realR; cwp3realF and cwp3realR; myb2realF and myb2realR; ranrealF and ranrealR; 18SrealF and 18SrealR [44].

### 4.6. Plasmid Construction

Synthetic oligonucleotides used are shown in Appendix A. All constructs were verified by DNA sequencing as previously described [44]. Plasmid 5′Δ5N-Pac was a gift from Dr. Steven Singer and Dr. Theodore Nash [50]. Plasmid pgCas9 has been described previously [44]. To make construct pPMBF1, the mbf1 gene and its 300 bp of 5′- flanking region were amplified with oligonucleotides mbf1KF and mbf1MR, digested with KpnI and MluI, and cloned into KpnI and MluI digested pPop2NHA [71]. To make construct pPMBF1m1, pPMBF1m2, pPMBF1m3m, or pPMBF1m4, the *mbf1* gene was amplified using two primer pairs mbf1m1F, mbf1m2F, mbf1m3F, or mbf1m4F, and mbf1MR, and mbf1m1R, mbf1m2R, mbf1m3R, or mbf1m4R and mbf1KF. The two PCR products were purified and used as templates for a second PCR. The second PCR also included primers mbf1KF and mbf1MR, and the product was digested with KpnI and MluI and cloned into the KpnI and MluI digested pPop2NHA [71].

The 612-bp 5′-flanking region of the mbf1 gene was amplified with oligonucleotides MBF15HF and MBF15NR, digested with *Hind*III/*Nco*I and cloned into *Hind*III/*Nco*I digested 5′Δ5N-Pac, resulting in MBF15. The 407-bp 3′-flanking region of the *mbf1* gene was amplified with oligonucleotides MBF13XF and MBF1KR, digested with *Xba*I/*Kpn*I and cloned into *Xba*I/*Kpn*I digested MBF15, resulting in MBF153. We used gene synthesis services from IDT to obtain the fragment MBF1-guide. The NCBI Nucleotide Blast search was used to avoid the potential off-target effects of guide sequence. The mbf1-guide was digested with *Kpn*I/*Eco*RI and cloned into *Kpn*I/*Eco*RI digested MBF153, resulting in pMBF1td.

### 4.7. Transfection and Western Blot Analysis

Cells transfected with the pP series plasmids containing the pac gene were selected and maintained with 54 μg/mL (100 μM) of puromycin as previously described [50]. For CRISPR/Cas9 system, *Giardia* trophozoites were transfected with plasmid pMBF1td and pgCas9, and then selected in 100 μM puromycin as previously described [44]. The culture medium in the first replenishment contained 6 μM Scr7 and 100 μM puromycin as previously described [44]. The MBF1td stable transfectants were established after selection with puromycin. Stable transfectants were maintained at 100 μM puromycin and were further analyzed by Western blotting, or DNA/RNA extraction. The replacement of the mbf1 gene with the pac gene was confirmed by PCR and sequencing. The control is *G. lamblia* trophozoites transfected with double amounts of 5′Δ5N-Pac plasmid and selected with puromycin as previously described [44]. Puromycin was then removed from the medium for each stable cell line to obtain MBF1td –pu, and control –pu cell lines [44]. Subsequent analysis was performed after the removal of the drug for 1 month.

Western blots were probed with anti-V5-HRP (Invitrogen, city, if any state (abbr.), country), anti-HA monoclonal antibody (1/5000 in blocking buffer; MilliporeSigma, Burlington, MA, United States), anti-MBF1 (1/10,000 in blocking buffer) (see below in this paper), anti-CWP1 (1/10,000 in blocking buffer) [20], anti-Myb2 (1/5000 in blocking buffer) [23], anti-α-tubulin (1/10,000 in blocking buffer, Sigma), anti-ISCS (1/10,000 in blocking buffer) [57], anti-RAN (1/10,000 in blocking buffer) [24], or preimmune serum (1/5000 in blocking buffer), and detected with HRP-conjugated goat anti-mouse IgG (1/5000; Thermo Fisher Scientific, Waltham, MA, USA) or HRP-conjugated goat anti-rabbit IgG (1/5000; Pierce) and enhanced chemiluminescence (Merck Millipore, MilliporeSigma, Burlington, MA, USA).

### 4.8. Expression and Purification of Recombinant MBF1 Protein

The genomic mbf1 gene was amplified using oligonucleotides MBF1F and MBF1R. The product was cloned into the expression vector pET101/D-TOPO (Invitrogen) in frame with the C-terminal His and V5 tags to generate plasmid pMBF1. To make the pMBF1m1, pMBF1m2, pMBF1m3, or pMBF1m4 expression vector, the *mbf1* gene was amplified using primers mbf1F and mbf1R and specific template, including pPMBF1m1, pPMBF1m2, pPMBF1m3, or pPMBF1m4, and cloned into the expression vector. The pMBF1m1, pMBF1m2, pMBF1m3, or pPMBF1m4 plasmid was transformed into *Escherichia coli* and purified as previously described [21]. Protein purity and concentration were estimated by Coomassie Blue and silver staining compared with serum albumin. MBF1, MBF1m1, MBF1m2, MBF1m3, or MBF1m4 was purified to apparent homogeneity (>95%).

### 4.9. Generation of Anti-MBF1 Antibody

Rabbit anti-MBF1 polyclonal antibody was generated using purified MBF1 protein through a commercial vendor (Angene, Taipei, Taiwan).

### 4.10. Immunofluorescence Assay

The pPMBF1, pPMBF1m1, pPMBF1m2, pPMBF1m3, or pPMBF1m4 stable transfectants were cultured in growth medium under puromycin selection. Cells cultured in growth medium or encystation medium for 24 h were harvested and subjected to immunofluorescence assay as previously described [11]. Anti-HA monoclonal antibody (1/300 in blocking buffer; Thermo Fisher Scientific, Waltham, MA, USA) and anti-mouse ALEXA 488 (1/500 in blocking buffer, Molecular Probes) were used as the detector.

### 4.11. Electrophoretic Mobility Shift Assay

Double-stranded oligonucleotides were 5′-end-labeled as previously described [52]. Binding reaction mixtures contained the components as previously described [19]. For reaction, 5 ng of purified MBF1, MBF1m1, MBF1m2, MBF1m3, or MBF1m4 protein was used. Competition and supershift assays were performed as previously described [19]. The protein-DNA complexes were resolved by a 6% polyacrylamide gel electrophoresis.

### 4.12. Co-Immunoprecipitation Assay

The stable cell lines were cultured in growth medium or inoculated into encystation medium with puromycin and harvested and lysed after 24 h as previously described [57]. The cell lysates were incubated with anti-HA antibody conjugated to beads as previously described [57]. For reciprocal immunoprecipitation experiments, anti-MBF1 was used to do immunoprecipitation. The lysates were incubated with 2 μg of anti-MBF1 antibody or preimmune serum for 2 h and then incubated with protein G plus/protein A-agarose (Merck) for 1 h. Proteins from the beads were analyzed by Western blotting using anti-HA monoclonal antibody (1/5000 in blocking buffer; Sigma), or anti-MBF1, anti-Myb2 (1/5000 in blocking buffer) [23], anti-ISCS (1/10,000 in blocking buffer) [57] as previously described [57].

### 4.13. ChIP Assays

The WB clone C6 cells cultured in growth medium were inoculated into encystation medium and harvested after 24 h and the assay was performed as previously described [20,57]. The precleared lysate were incubated with 2 μg of anti-MBF1 or preimmune serum and then incubated with protein G plus/protein A-agarose (Merck) as previously described [57]. The beads were washed and then incubated with elution buffer as previously described [57]. DNA was purified and subjected to PCR reaction followed by agarose gel electrophoresis or to quantitative real-time PCR as previously described [57]. Primers 18S5F and 18S5R were used to amplify the *18S ribosomal RNA* gene promoter as a control for our ChIP analysis. Primers mbf15F and mbf15R, cwp15F and cwp15R, cwp25F and cwp25R, cwp35F and cwp35R, myb25F and myb25R, U65F and U65R were used to amplify mbf1, cwp1, cwp2, cwp3, myb2, and U6 gene promoters within the −200 to −1 region.

### 4.14. Microarray Analysis

Microarray analysis was performed as previously described using RNA from the pPMBF1 (labeled by Cy5) and 5′Δ5N-Pac cell lines (labeled by Cy3) [57]. All data is MIAME compliant and that the raw data have been deposited in a MIAME (http://www.mged.org/Workgroups/MIAME/miame.html) compliant database (GEO) with accession number GSE162220.

## Figures and Tables

**Figure 1 ijms-22-01370-f001:**
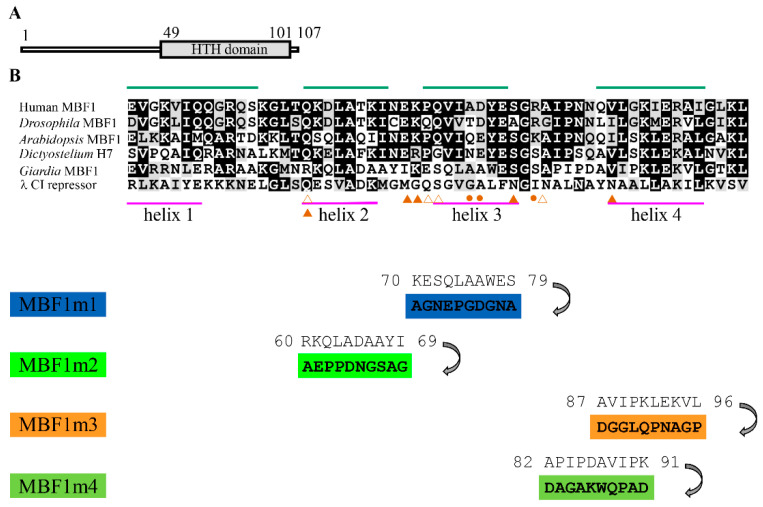
Domain architecture of multiprotein bridging factor 1 (MBF1) protein and alignment of the HTH domains. (**A**) Schematic representation of the *Giardia* MBF1 protein. The gray box indicates the helix-turn-helix (HTH) domain, as predicted by pfam. (**B**) Alignment of various HTH domains and sequence of *Giardia* MBF1 mutants. The MBF1 proteins from different organisms, including human, *Drosophila*, *Arabidopsis*, *Dictyostelium* and *Giardia*, are analyzed by ClustalW 1.83 with all default settings. GenBank *accession* numbers for human MBF1, *Drosophila* MBF1, *Arabidopsis* MBF1, *Dictyostelium* H7, *Giardia* MBF1, and λ CI repressor are NP_003783.1, NP_524110.1, NP_565981.1, CAA33444.1, XP_001704109.1, and AAK07889.1, respectively. Black and gray boxes indicate identical and conserved amino acids in the respective proteins, respectively. The predicted 4 α helixes of HTH domain of *Giardia* MBF1 and λ CI repressor are indicated by green and purple lines, respectively. The helix 2 and helix 3 are the core structure of λ CI repressor and the latter is the recognition helix. Amino acids of λ CI repressor that contact DNA major groove and DNA phosphates are indicated by filled and open triangles, respectively. The hydrophobic amino acids of λ CI repressor that increase DNA binding specificity are indicated by filled circles. MBF1m1 contains mutation of residues 70–79 corresponding to helix 3. MBF1m2 contains mutation of residues 60–69 corresponding to helix 2. MBF1m3 contains mutation of residues 87–96 corresponding to helix 4. MBF1m4 contains mutation of residues 82–91 corresponding to the turn between helix 3 and helix 4 and part of helix 4.

**Figure 2 ijms-22-01370-f002:**
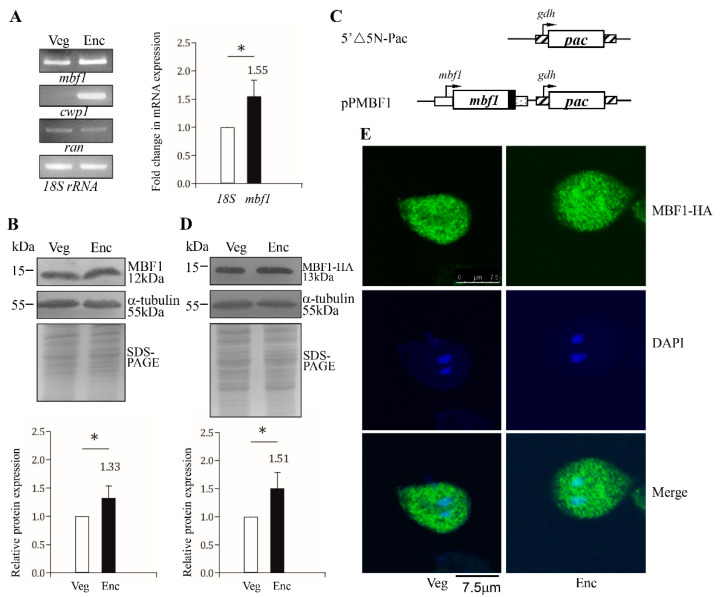
MBF1 protein and mRNA expression. (**A**) RT-PCR and quantitative real-time PCR analysis. RNA was extracted from *G. lamblia* wild type non-transfected WB cells cultured in growth medium (Veg, vegetative growth) or encystation medium for 24 h (Enc, encystation). RT-PCR was performed using primers for genes indicated in the figure (left panel). The mRNA levels of the cwp1 and ran genes significantly increased and decreased during encystation, respectively. Real-time PCR was performed using primers for genes indicated in the figure (right panel). The mRNA levels were normalized to the 18S ribosomal RNA level. The ratio of mRNA levels in Enc sample to levels in Veg sample is shown and expressed as the means ± 95% confidence intervals of at least three separate experiments. *, *p* < 0.05. (**B**) MBF1 protein level increased during encystation. The wild type non-transfected WB cells were incubated in growth medium (Veg, vegetative growth) or encystation medium for 24 h (Enc, encystation) and then subjected to Western blot analysis using anti-MBF1 and anti-α-tubulin antibodies, respectively. SDS-PAGE and Coomassie Blue staining is included as a control for equal protein loading. The level of α-tubulin protein is slightly decreased in encystation sample. The band intensity from triplicate Western blots was quantified using Image J. The MBF1 protein levels were normalized to the loading control (Coomassie Blue-stained proteins). The ratio of MBF1 protein levels in Enc sample to levels in Veg sample is shown and expressed as mean ± 95% confidence intervals. *, *p* < 0.05. (**C**) Schematic presentation of the 5′Δ5N-Pac and pPMBF1 plasmid. The pac gene (open box) is flanked by the 5′- and 3′-untranslated regions of the glutamate dehydrogenase (gdh) gene (striated box). In construct pPMBF1, the mbf1 gene is flanked by its own 5′-untranslated region (open box) and the 3′-untranslated region of the ran gene (dotted box). The coding sequence of the HA epitope tag is shown as a filled black box. (**D**) MBF1-HA protein level increased in MBF1-overexpressing cells during encystation. The pPMBF1 stable transfectants were incubated in growth medium (Veg, vegetative growth) or encystation medium for 24 h (Enc, encystation) and then subjected to Western blot analysis using anti-HA and anti-α-tubulin antibodies, respectively. SDS-PAGE with Coomassie Blue staining is included as a control for equal protein loading. The level of α-tubulin protein was slightly decreased in encystation sample. The band intensity and the ratio of MBF1-HA protein levels in Enc sample to levels in Veg sample is calculated as described in Figure 1B. (**E**) Localization of MBF1 protein. The pPMBF1 stable transfectants were incubated in growth medium (Veg, left panel) or encystation medium for 24 h (Enc, right panel), and immunofluorescence assays were performed using anti-HA antibody. The upper panels show the localization of the MBF1 protein (green). The middle and bottom panels show the DAPI staining of cell nuclei (blue) and the merged images, respectively.

**Figure 3 ijms-22-01370-f003:**
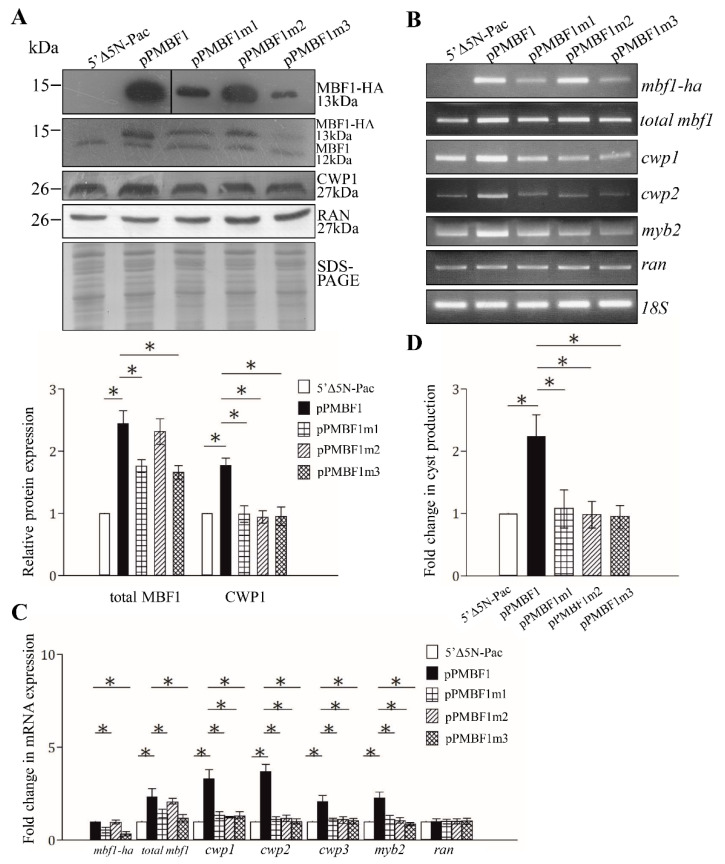
Increased expression of the cwp1-3 and myb2 genes in the MBF1-overexpressing cell line. (**A**) MBF1 overexpression increased the cyst wall protein 1 (CWP1) level. The mbf1 gene was mutated (Figure 1B) and subcloned to replace the wild-type mbf1 gene in the backbone of pPMBF1 (Figure 2C), and the resulting plasmids pPMBF1m1-3 were transfected into *Giardia*. The 5′∆5N-Pac, pPMBF1, and pPMBF1m1-3 stable transfectants were incubated in growth medium and then subjected to Western blot analysis using anti-HA, anti-MBF1, anti-CWP1, and anti-Ran antibodies, respectively. SDS-PAGE with Coomassie Blue staining is included as a control for equal protein loading. As a control, similar levels of the RAN protein were detected. The band intensity from triplicate Western blots was quantified using Image J. The MBF1 or CWP1 protein levels were normalized to the RAN loading control. The ratio of MBF1 or CWP1 protein levels in the specific cell line to levels in the 5′∆5N-Pac cell line is shown and expressed as mean ± 95% confidence intervals. *, *p* < 0.05. (**B**) RT-PCR assays of transcript expression in the MBF1- and MBF1 mutants- expressing cell lines. RNA was extracted from the 5′∆5N-Pac, pPMBF1, and pPMBF1m1-3 stable transfectants cultured in growth medium. RT-PCR was performed using primers for genes indicated in the figure. As a control, similar levels of the 18S ribosomal RNA (18S) were detected. (**C**) Quantitative real-time PCR assays of transcript expression in the MBF1- and MBF1 mutants- expressing cell lines. Real-time PCR was performed using primers for genes indicated in the figure. Similar levels of the 18S ribosomal RNA were detected. The mRNA levels were normalized to the 18S ribosomal RNA levels. The ratio of mRNA levels in the pPMBF1 or pPMBF1m1-3 cell line to levels in the 5′∆5NPac cell line is shown and expressed as the mean ± 95% confidence intervals of at least three separate experiments. *, *p* < 0.05. (**D**) Overexpression of MBF1 induced cyst generation. Cyst number was counted from the 5′∆5N-Pac, pPMBF1, and pPMBF1m1-3 stable transfectants cultured in growth medium as described in “Materials and Methods”. Fold changes in cyst generation are shown as the ratio of the sum of total cysts in the pPMBF1 or pPMBF1m1-3 cell lines relative to the 5′∆5NPac cell line. Values are shown as mean ± 95% confidence intervals. *, *p* < 0.05.

**Figure 4 ijms-22-01370-f004:**
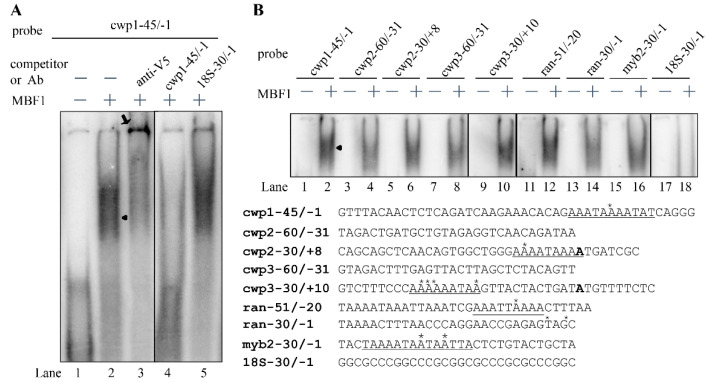
DNA-binding ability of MBF1 revealed by electrophoretic mobility shift assays. (**A**) Detection of MBF1 binding sites. Purified MBF1 and the ^32^P-end-lableled oligonucleotide probe cwp1-45/-1 (-45 to -1 relative to the translation start site of the cwp1 gene) were used in electrophoretic mobility shift assays. Binding reaction mixtures contained the components indicated above the lanes. For competition and supershift assays, some reaction mixtures contained 200-fold molar excess of cold oligonucleotides cwp1-45/-1 or 18S-30/-1 or 0.8 μg of anti-V5-horseradish peroxidase (HRP) antibody. The arrowhead and arrow indicate the shifted complex and supershifted band, respectively. (**B**) MBF1 binding sites in various promoters. Purified MBF1 and various ^32^P-end-lableled oligonucleotide probes as indicated above the lanes were used in electrophoretic mobility shift assays. The arrowhead indicates the shifted complex. The transcription start sites of the cwp1-3 and myb2 genes in 24-h encysting cells are indicated by asterisks. The transcription start sites of the ran gene determined from vegetative cells are indicated by asterisks. The transcription start sites are within the AT-rich initiator (Inr) elements, which are underlined. The translation start sites of the cwp2 and cwp3 genes are in bold. “18S” represents 18S ribosomal RNA.

**Figure 5 ijms-22-01370-f005:**
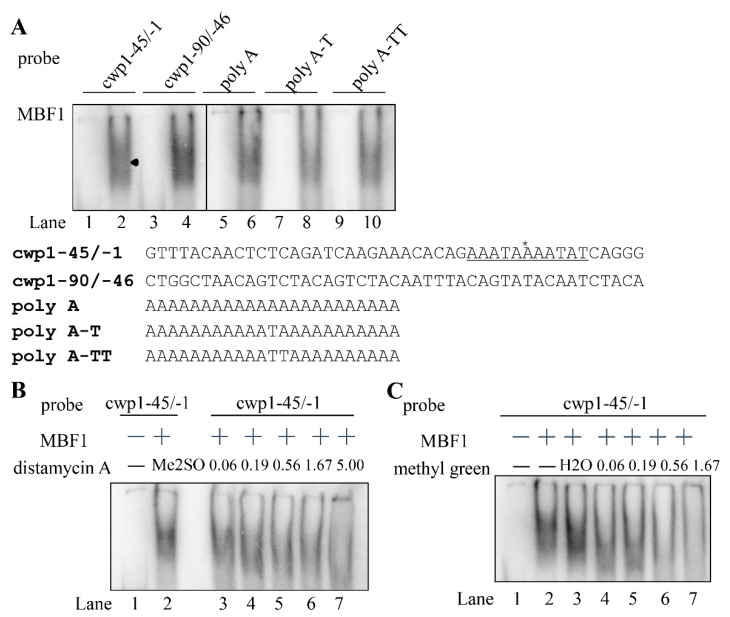
Effect of DNA binding drugs on DNA binding ability of MBF1. (**A**) MBF1 may bind to AT-rich sequence. Purified MBF1 and various ^32^P-end-lableled oligonucleotide probes as indicated above the lanes were used in electrophoretic mobility shift assays. The arrowhead indicates the shifted complex. The transcription start site and AT-rich initiator (Inr) element of the cwp1 gene in 24-h encysting cells is indicated by asterisks and underlined letters, respectively. Effect of distamycin A (**B**) or methyl green (**C**) on the binding of MBF1 to DNA. ^32^P end-labeled cwp1-45/-1 probe was incubated with MBF1 in the presence of (**B**) distamycin A or (**C**) methyl green (lanes 3–7) using the same volume of (**B**) Me_2_SO or (**C**) H_2_O as a solvent control.

**Figure 6 ijms-22-01370-f006:**
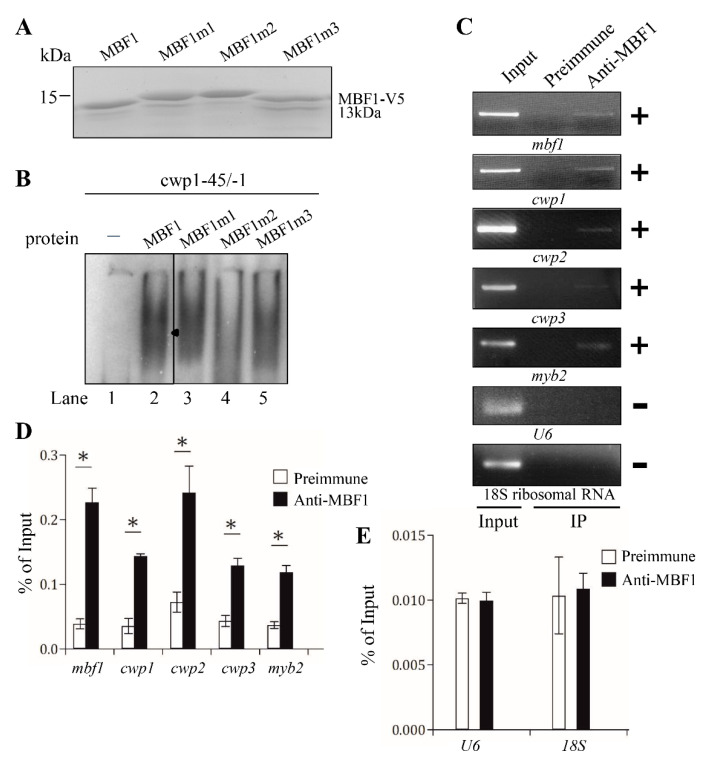
Analysis of the DNA-binding domain of MBF1. (**A**) SDS-PAGE analysis of recombinant MBF1 and MBF1m1-3 proteins. MBF1 or MBF1m1-3 proteins with a C-terminal V5 tag were purified from *E. coli* by affinity chromatography and then analyzed by SDS-PAGE and silver staining. (**B**) Decreased DNA-binding ability of MBF1m2. Purified MBF1 and its mutants and the ^32^P-end-labeled oligonucleotide probe cwp1-45/-1 were used in electrophoretic mobility shift assays. The arrowhead indicates the shifted complex. (**C**) Recruitment of MBF1 to the cwp1-3 and myb2 promoters. The non-transfected WB cells were cultured in encystation medium for 24 h and then subjected to chromatin immunoprecipitation (ChIP) assays. Anti-MBF1 antibody was used to assess interaction of MBF1 to evarious gene promoters. Preimmune serum was used as a negative control. Immunoprecipitated chromatin was analyzed by PCR using primers that amplify the 5′-flanking region of the genes indicated in the figure. The anti-MBF1 antibody enriched the mbf1, cwp1, cwp2, cwp3, and myb2 promoter fragments (+), but not the U6 promoter fragment (−). The *18S ribosomal RNA* gene promoter was used as a negative control (−). (**D**,**E**) ChIP assays coupled with quantitative PCR. Values represented as a percentage of the antibody-enriched chromatin relative to the total input chromatin (% of Input). Results are expressed as the mean ± 95% confidence intervals of at least three experiments. *, *p* < 0.05.

**Figure 7 ijms-22-01370-f007:**
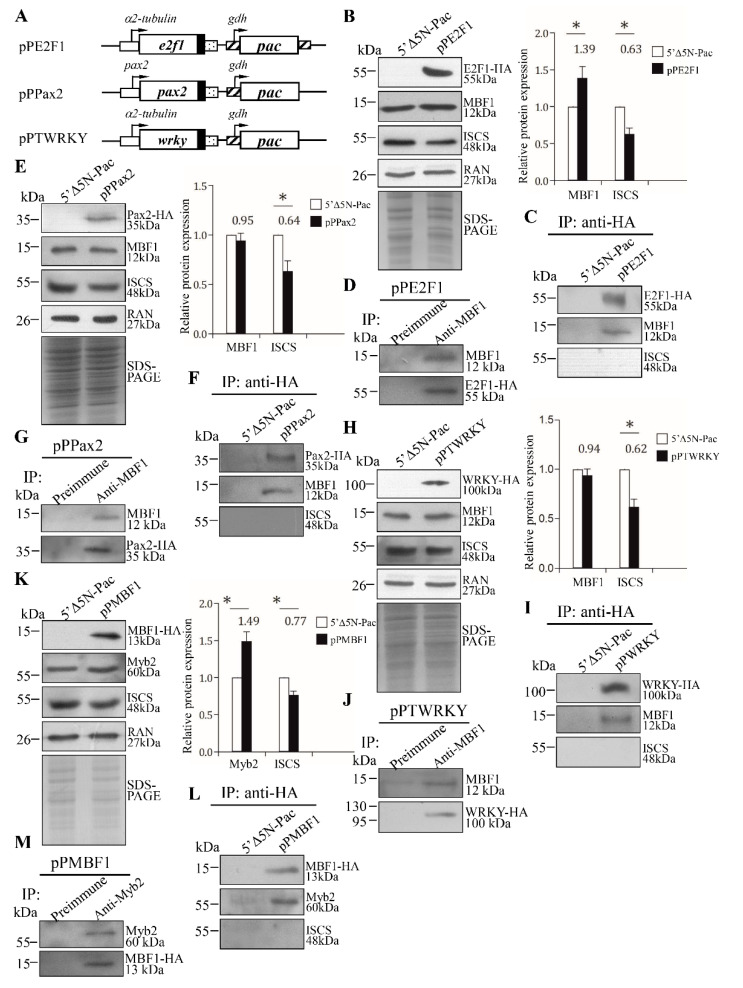
Interaction between MBF1 and E2F1, Pax2, WRKY, or Myb2. (**A**) Diagrams of the pPE2F1, pPPax2, or pPTWRKY plasmid. The expression cassette of the pac gene and the HA epitope tag coding sequence are the same as in Figure 2C. The e2f1, pax2, or wrky gene is under the control of α-tubulin promoter or its own 5′-flanking region (open box) and the 3′-flanking region of the ran gene (dotted box). Expression of the HA-tagged (**B**) E2F1, (**E**) Pax2, or (**H**) WRKY protein in whole cell extracts for co-immunoprecipitation assays (Input). The 5′∆5N-Pac and (**B**) pPE2F1, (**E**) pPPax2, or (**H**) pPTWRKY stable transfectants were cultured in encystation medium for 24 h. SDS-PAGE and Western blot were performed as described in Figure 3A. The blot was probed with anti-HA, anti-MBF1, anti-ISCS, and anti-RAN antibodies, respectively. The band intensity from triplicate Western blots was quantified using Image J as described in Figure 3A. *, *p* < 0.05. Interaction between MBF1 and (**C**) E2F1, (**F**) Pax2, or (I) WRKY was detected by co-immunoprecipitation assays. The 5′∆5N-Pac and (**C**) pPE2F1, (**F**) pPPax2, or (**I**) pPTWRKY stable transfectants were cultured in encystation medium for 24 h. Proteins from cell lysates were immunoprecipitated using anti-HA antibody and analyzed by Western blotting with anti-HA, anti-MBF1, and anti-ISCS antibodies, respectively. Reciprocal immunoprecipitation for confirmation of MBF1 and (**D**) E2F1, (**G**) Pax2, or (**J**) WRKY interaction. The (**D**) pPE2F1, (**G**) pPPax2, or (**J**) pPTWRKY stable transfectants were cultured in encystation medium for 24 h. Proteins from cell lysates were immunoprecipitated using anti-MBF1 antibody or preimmune serum (negative control) and analyzed by Western blotting with anti-MBF1 and anti-HA antibodies, respectively. (**K**) Expression of the HA-tagged MBF1 protein in whole cell extracts for co-immunoprecipitation assays (Input). The 5′∆5N-Pac and pPMBF1 stable transfectants were cultured in encystation medium for 24 h. SDS-PAGE and Western blot were performed as described in Figure 3A. The blot was probed with anti-HA, anti-Myb2, anti-ISCS, and anti-RAN antibodies, respectively. The band intensity from triplicate Western blots was quantified using Image J as described in Figure 3A. *, *p* < 0.05. (**L**) Interaction between MBF1 and Myb2 was detected by co-immunoprecipitation assays. The 5′∆5N-Pac and pPMBF1 stable transfectants were cultured in encystation medium for 24 h. Proteins from cell lysates were immunoprecipitated using anti-HA antibody and analyzed by Western blotting with anti-HA, anti-Myb2, and anti-ISCS antibodies, respectively. (**M**) Reciprocal immunoprecipitation for confirmation of MBF1 and Myb2 interaction. The pPMBF1 stable transfectants were cultured in encystation medium for 24 h. Proteins from cell lysates were immunoprecipitated using anti-Myb2 antibody or preimmune serum (negative control) and analyzed by Western blotting with anti-Myb2 and anti-HA antibodies, respectively.

**Figure 8 ijms-22-01370-f008:**
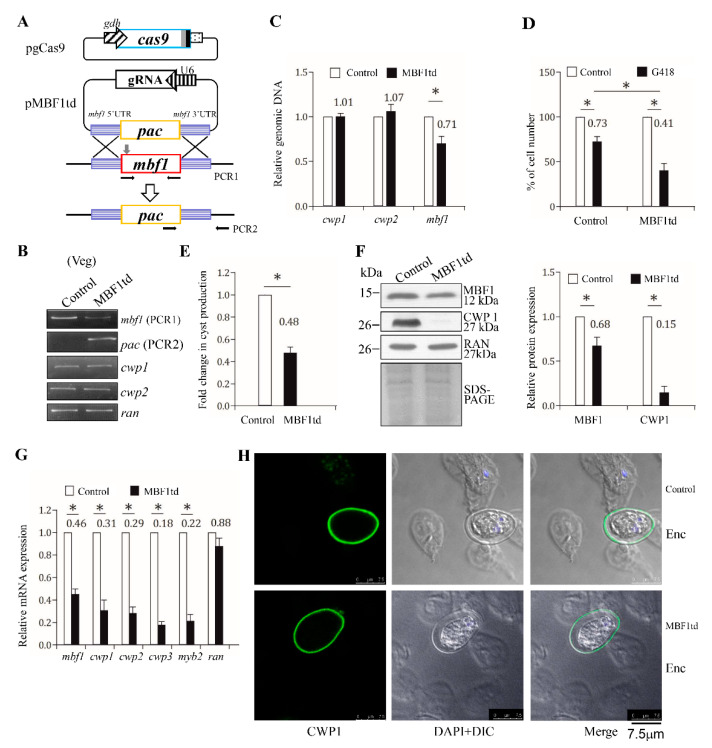
Targeted disruption of the mbf1 gene resulted in decreased expression of the cwp1-3 and myb2 genes during vegetative growth. (**A**) Schematic presentation of the pgCas9 and pMBF1td plasmids. In construct pgCas9, the cas9 gene is flanked by gdh promoter (striated box) and 3′ untranslated region of the ran gene (dotted box). The nuclear localization signal (filled gray box) and an HA tag (filled black box) are fused to the C terminus. In construct pMBF1td, a single gRNA is under the control of the *Giardia* U6 promoter. The single gRNA, which is located upstream three nucleotides of protospacer-adjacent motif (NGG sequence), includes a guide sequence targeting 20-nucleotide of the mbf1 gene (nt 126–145). pMBF1td also has the HR template cassette composed of the pac selectable marker and the 5′ and 3′ flanking region of the mbf1 gene as homologous arms. The Cas9/gRNA cutting site in the genomic mbf1 gene is indicated by a gray arrow. Replacement of the genomic mbf1 gene with the pac gene will occur by HR, after introducing a double-stranded DNA break in the mbf1 gene. After transfection of the pgCas9 and pMBF1td constructs into *G. lamblia* WB trophozoites, the MBF1td stable transfectants were established under puromycin selection. The control cell line is trophozoites transfected with double amounts of 5′Δ5N-Pac plasmid (Figure 2C) and selected with puromycin. PCR1/2 were used for identification of clones with targeted disruption. (**B**) PCR confirmed partial replacement of the mbf1 gene with the pac gene in the MBF1td cell line. Genomic DNA was isolated from the MBF1td and control cell lines cultured in growth medium with puromycin (vegetative growth, Veg) and subjected to PCR using primers specific for mbf1 (PCR1 in panel A), pac (PCR2 in panel A), cwp1, cwp2, and ran genes, respectively. Products from the cwp1, cwp2, and ran genes are internal controls. (**C**) real-time PCR confirmed partial disruption of the mbf1 gene in the MBF1td cell line. Real-time PCR was performed using primers specific for mbf1, cwp1, cwp2, and ran genes, respectively. The mbf1, cwp1, and cwp2 DNA levels were normalized to the ran DNA level. The ratio of DNA levels in MBF1td cell line to levels in control cell line is shown and expressed as the means ± 95% confidence intervals of at least three separate experiments. *, *p* < 0.05. (**D**) G418 sensitivity increased by targeted disruption of the mbf1 gene. The MBF1td and control cell lines were subcultured in growth medium containing 518 μM G418 for 24 h and then subjected to cell count. An equal volume of ddH2O was added to cultures as a negative control. Fold changes in cell number are shown as the ratio of cell number in the G418 sample relative to the ddH2O sample. Values are shown as mean ± 95% confidence intervals of three independent experiments. *, *p* < 0.05. (**E**) Targeted disruption of the mbf1 gene in the MBF1td cell line resulted in decreased cyst generation during vegetative growth. Cyst number was counted from the control and MBF1td cell lines cultured in growth medium as described in “Materials and Methods” and Figure 3D. *, *p* < 0.05. (**F**) The CWP1 level decreased by targeted disruption of the mbf1 gene in the MBF1td cell line during vegetative growth. The control and MBF1td cell lines cultured in growth mediumwere subjected to SDS-PAGE and Western blot analysis as described in Figure 3A. The blot was probed with anti-MBF1, anti-CWP1, and anti-RAN antibodies, respectively. The band intensity from triplicate Western blots was quantified using Image J as described in Figure 3A. *, *p* < 0.05. (**G**) Targeted disruption of the mbf1 gene in the MBF1td cell line resulted in decreased expression of cwp1-3 and myb2 during vegetative growth. The control and MBF1td cell lines cultured in growth medium were subjected to quantitative real-time RT-PCR analysis using primers specific for mbf1, cwp1, cwp2, cwp3, myb2, ran, and 18S ribosomal RNA genes, respectively, as described in Figure 2A. *, *p* < 0.05. (**H**) Cyst wall change by targeted disruption of the mbf1 gene. The control and MBF1td cell lines with puromycin selection were cultured in encystation medium for 24 h and then subjected to immunofluorescence assay. The endogenous CWP1 protein was detected by anti-CWP1 antibody. The left panel shows the localization of the CWP1 protein with green-stained cyst wall. The middle panel shows the merge of DAPI (blue) and differential interference contrast images. The right panel shows the merged images. The intensity and thickness of cyst wall from immunofluorescence assays was quantified using Image J. The average thickness of cyst wall of the control and MBF1td cell lines was estimated to be 421 nm and 263 nm, respectively. (*n* = 15–20 cysts/condition).

**Figure 9 ijms-22-01370-f009:**
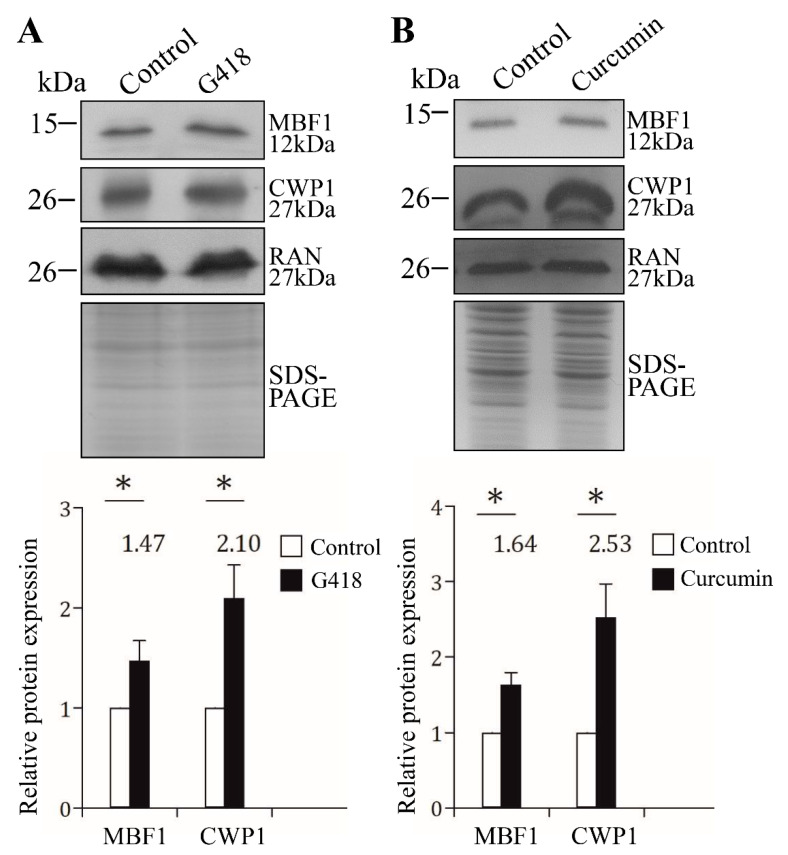
G418 and curcumin treatment increased the levels of MBF1 and CWP1 proteins. The wild-type non-transfected WB cells were cultured in growth medium containing (**A**) 217 µM G418 or (**B**) 200 µM curcumin for 24 h using the same volume of (**A**) H_2_O or (**B**) Me_2_SO as a solvent control and then subjected to SDS-PAGE and Western blot analysis. The blot was probed with anti-MBF1, anti-CWP1, and anti-RAN antibodies, respectively. The intensity of bands was quantified as described in Figure 2B. *, *p* < 0.05.

**Figure 10 ijms-22-01370-f010:**
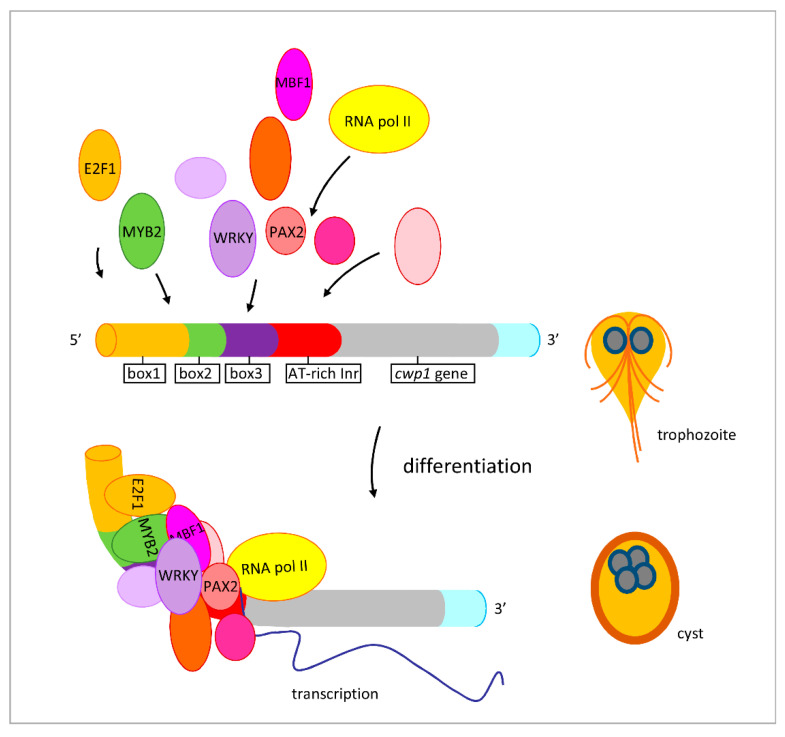
Increased expression of encystation-induced cwp1 gene in cyst differentiation process. One gene encoding key construction component of the cyst wall, cwp1, is up-regulated by MBF1, E2F1, Pax2, WRKY, Myb2, and other transcription factors in cyst differentiation process. These transcription factors can bind to cis-acting elements of the cwp1 promoter, such as box1-3 or AT-rich initiator (Inr), to activate cwp1 gene transcription. They can form complexes to recruit RNA polymerase II for transcription initiation. CWP1 is present in vegetative trophozoite stage at a lower level and displays higher expression levels by these transcription factors in cyst differentiation process. The increase of MBF1, E2F1, Pax2, WRKY, Myb2, and other transcription factors during encystation may further induce CWP1 expression, resulting in more cyst generation.

## Data Availability

Publicly available datasets were analyzed in this study. This data can be found here: https://www.ncbi.nlm.nih.gov/geo/query/acc.cgi?acc=GSE162220.

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
