# Peer review of "A Novel Multiprotein Bridging Factor 1-Like Protein Induces Cyst Wall Protein Gene Expression and Cyst Differentiation in Giardia lamblia"

_ijms, 2021, doi:10.3390/ijms22031370_

Round 1

Reviewer 1 Report

The cell biology associated with encystation in Giardia is the focus of much work and over the past few years, the machinery associate with this process has been slowly uncovered. The work presented is very compelling and utilises a wide range of technologies. The authors have shown a link between MBF1 and encystation and the knockdown down data suggests that there is indeed a reduction in cyst formation.  I do have a few comments re the data and presentation

  • With respect to encystation, the data is presented as fold decrease/ increase – this is g=fine however it would be good to have figures showing the percentage cyst formation in the control and knockout groups. Although relatively trivial, it does indicate the clear impact of the treatment on cyst formation. In addition the level of cysts formation in low bile media would be good to include in this as it sets the bar for the process
  • From the data above, the impact of this knockout and the significance of the change can be better assessed and this in part links to Figure 10. I like this diagram however I think it could be enhanced by suggesting the interaction of the other TF’s - how significant/ important do the authors think  MBF1 is in the overall process?
  • Did the authors observe differences in the time taken to observe cysts in the MBF1td cell line by comparison to the control?
  • A minor point is that the thickness of the cell wall is reduced in MBF1td cell line – how was that determined
  • I realise that some of the techniques are difficult with respect to output however the quality of the images in figure 4 are poor by comparison to the others presented are there better images that could be used?

Author Response

We have revised this manuscript according to the critiques and questions of reviewers. The responses and the changes in the text are detailed below each critique.

The cell biology associated with encystation in Giardia is the focus of much work and over the past few years, the machinery associate with this process has been slowly uncovered. The work presented is very compelling and utilises a wide range of technologies. The authors have shown a link between MBF1 and encystation and the knockdown down data suggests that there is indeed a reduction in cyst formation.  I do have a few comments re the data and presentation

With respect to encystation, the data is presented as fold decrease/ increase – this is g=fine however it would be good to have figures showing the percentage cyst formation in the control and knockout groups. Although relatively trivial, it does indicate the clear impact of the treatment on cyst formation.

Response:We did not determine the percentage cyst formation in these experiments. However, we followed the same protocol with high bile medium for encystation from Dr. Gillin’s lab. According to Dr. Gillin, encystation is generally at least 80% of the population for wild-type Giardia WB clone (personal communication).

In addition the level of cysts formation in low bile media would be good to include in this as it sets the bar for the process

Response:We used the most updated protocol with high bile medium for encystation in Dr. Gillin’s lab (please see ref 17, Sun et al., 2002, in the Materials and Methods section: “Giardia trophozoites were collected from growth medium and encysted for the times indicated in TYI-S-33 medium with 12.5 mg ml-1 of bovine bile at pH 7.8 at a beginning density of 5 x10^5 cells ml-1.”) The protocol with low bile medium was not used in our lab, so we did not test with the low bile medium from the beginning.

From the data above, the impact of this knockout and the significance of the change can be better assessed and this in part links to Figure 10. I like this diagram however I think it could be enhanced by suggesting the interaction of the other TF’s - how significant/ important do the authors think  MBF1 is in the overall process?

Response:This is a good point. We added “It is possible that interaction of MBF1 with some of these transcription factors can stabilize their binding to DNA. A similar phenomenon was reported for the MBF1–FTZ-F1 interaction in silkworm [35].” Page 17.

Did the authors observe differences in the time taken to observe cysts in the MBF1td cell line by comparison to the control?

Response:We did not count the number of cysts in these experiments over all time course study. We show the data of cyst number at 24h encystation, which is the time with maximal expression of cyst wall protein (ref16, Sun et al., 2003).

A minor point is that the thickness of the cell wall is reduced in MBF1td cell line – how was that determined

Response:We used image J to measure the intensity and thickness of cyst wall. We added “The intensity and thickness of cyst wall from immunofluorescence assays was quantified using Image J. The average thickness of cyst wall of the control and MBF1td cell lines was estimated to be 421nm and 263nm, respectively. (n = 15-20 cysts/condition).” Page 15.

I realise that some of the techniques are difficult with respect to output however the quality of the images in figure 4 are poor by comparison to the others presented are there better images that could be used?

Response:In Fig. 4, the EMSA experiments were performed with harmful radioisotope reagent, and we tried to present the best image for our manuscript.

Reviewer 2 Report

Article entitled "A novel multiprotein bridging factor 1-like protein induces cyst wall protein gene expression and cyst differentiation in Giardia lamblia " is an interesting publication especially for researchers and physicians from regions with high morbidity and mortality caused by giardiasis. The described findings might be also useful during the realisation of projects aimed to prepare novel therapies or vaccines against Giardia lamblia infections. Therefore I suggest to accept this paper in its present form.

Author Response

Dear IJMS Editor,

We have revised this manuscript according to the critiques and questions of reviewers. The responses and the changes in the text are detailed below each critique. They are marked by //.

We hope that you will now fine it appropriate for publication in IJMS.

Thank you very much.

Sincerely,

Chin-Hung Sun